# Live visualisation of electrolytes during mouse embryonic development using electrolyte indicators

**Akiko Fujishima** [1]*, **Kazumasa Takahashi**[1], **Mayumi Goto**[1], **Takeo Hirakawa**[1], **Takuya Iwasawa**[2], **Kazue Togashi**[1], **Eri Maeda**[3], **Hiromitsu Shirasawa**[1], **Hiroshi Miura**[1], **Wataru Sato**[1], **Yukiyo Kumazawa**[1], **Yukihiro Terada**[1]

1 Department of Obstetrics and Gynecology, Akita University Graduate School of Medicine, Akita, Japan, 2 Department of Obstetrics and Gynecology, Omagari Kousei Medical Center, Akita, Japan, 3 Department of Environmental Health Science and Public Health, Akita University Graduate School of Medicine, Akita, Japan

* fujishimaa@med.akita-u.ac.jp

**Data Availability Statement:** All data generated or analysed during this study are included in this published article.

## Abstract

Studies have shown that some electrolytes, including $Na^+$ and $K^+$, play important roles in embryonic development. However, these studies evaluated these electrolytes by using inhibitors or knockout mice, with no mention on the changes in the intracellular electrolyte concentrations during embryogenesis. In this study, we used the electrolyte indicators CoroNa Green AM and ION Potassium Green-2 AM to directly visualise intracellular concentrations of $Na^+$ and $K^+$, respectively, at each embryonic developmental stage in mouse embryos. We directly observed intracellular electrolyte concentrations at the morula, blastocyst, and hatching stages. Our results revealed dynamic changes in intracellular electrolyte concentrations; we found that the intracellular $Na^+$ concentration decreased, while $K^+$ concentration increased during blastocoel formation. The degree of change in intensity in response to ouabain, an inhibitor of $Na^+/K^+$ ATPase, was considered to correspond to the degree of $Na^+/K^+$ ATPase activity at each developmental stage. Additionally, after the blastocyst stage, trophectoderm cells in direct contact with the blastocoel showed higher $K^+$ concentrations than in direct contact with inner cell mass, indicating that $Na^+/K^+$ ATPase activity differs depending on the location in the trophectoderm. This is the first study to use CoroNa Green AM and ION Potassium Green-2 AM in mouse embryos and visualise electrolytes during embryonic development. The changes in electrolyte concentration observed in this study were consistent with the activity of $Na^+/K^+$ ATPase reported previously, and it was possible to image more detailed electrolyte behaviour in embryo cells. This method can be used to improve the understanding of cell physiology and is useful for future embryonic development studies.

## Introduction

Pre-implantation development in mouse embryo involves sequential division of a fertilised egg into blastomeres. After the 8-cell stage, blastomeres undergo compaction leading to the formation of a morula, which is the earliest point of differential spatial positioning, where the cells

**Funding:** This work was supported by Japan Society for the Promotion of Science, Grant-in-Aid for Scientific Research (B) [Grant Number 18H02942].

**Competing interests:** The authors have declared that no competing interests exist.

are polarised along the apical-basal axis [1]. The embryo then forms the blastocyst, characterised by the presence of a fluid-filled cavity and inner cell mass (ICM) that are both surrounded by the trophectoderm (TE) [2].

Many researchers have reported electrolyte involvement in embryo development [3]. For example, at fertilisation, $Ca^{2+}$ oscillation is required for the initiation of embryonic development [4], and $K^+$ is thought to be involved in cell cycle regulation [5]. Particularly, many studies examining blastocoel formation have reported that water movement, along with the $Na^+$ concentration gradient, $Na^+/H^+$ exchangers, and $Na^+$ channel, play key roles in $Na^+$ influx at the apical membrane [6]. Additionally, $Na^+/K^+$ ATPase plays a major role in $Na^+$ efflux at the blastocoel membrane [7–9]. Water channels, known as aquaporins, are essential to facilitate water influx through the outer cell [7, 9].

These studies primarily used knockout mice to evaluate each channel or inhibitor. To date, no studies have directly assessed intracellular ion concentrations after the morula stage. Because blastocoel formation is the most dynamic process in embryo development, it is important to elucidate electrolyte behaviours at these stages.

Furthermore, live visualisation of $Na^+$ and $K^+$ behaviour during embryonic development is necessary to advance the development of reproductive medicine. As previously mentioned, $Na^+$ is the most important electrolyte in blastocoel formation. Typically, the intracellular $Na^+$ concentration in mammals varies between 5–15 mM depending on the cell type, whereas the extracellular $Na^+$ concentration is much higher at approximately 120–150 mM [10]. This concentration gradient is established by the activity of ion channels/transporters and pumps and is required to maintain the resting membrane potential [10]. Moreover, $K^+$ plays an important role in embryonic development. Alterations in $K^+$ channel activity induce changes in membrane potential, which may modulate physiological functions at the cellular level such as gamete maturation, fertilisation, cell division, and early embryonic development [5]. $K^+$ channels are active in early mouse oocytes until at least the blastocyst stage, oscillating in phase with the developmental cell cycle, and exhibit high and low activities in the $M/G_1$ and $S/G_2$ phases, respectively [11]. Additionally, high expression of the K2P channel in the morula is likely to be a key factor in blastocyst formation [5].

In recent years, some studies have employed an electrolyte indicator to clarify real-time electrolyte behaviour in nerve fibres and cardiomyocytes [12, 13]. In the reproductive field, some studies used $Ca^{2+}$ indicators [14] but none have reported the use of $Na^+$ and $K^+$ electrolyte indicators. CoroNa Green AM and ION Potassium Green-2 AM, which are indicators for $Na^+$ and $K^+$, respectively, exhibit increased fluorescence emission intensity upon binding of $Na^+$ and $K^+$. These indicators possess cell-permeant acetoxymethyl (AM) esters. Once inside the cell, the AM esters are hydrolysed by intracellular esterases to release the dye. Thus, the electrolyte concentration in the cell can be quantified based on the fluorescence intensity.

In this study, we used these electrolyte indicators to visualise $Na^+$ and $K^+$ concentrations at each developmental stage in mouse embryos, to gain insights into the cytophysiology of embryonic development. Furthermore, immunostaining of $Na^+/K^+$ ATPase was performed to elucidate the relationship between $Na^+/K^+$ ATPase expression and activity at each developmental stage.

## Materials and methods

### Ethical approval

Approval for the animal studies was obtained from the Institutional Animal Care and Use Committee of Akita University (permission number: 1090–2). This study was conducted according to Akita University Animal Experimentation Regulations.

## Collection of mouse oocytes, zygotes, and embryos

All animal experiments were performed in accordance with the Guide for Care and Use of Laboratory Animals of Akita University and carried out as previously described with some modifications [15]. All mice were housed in cages at 23˚C with a 12-h light/dark photoperiod and free access to food and water. *In vitro* fertilisation (IVF) was selected as the embryo acquisition method to precisely determine the developmental stage and maintain a uniform developmental environment throughout the experimental period, starting from the time of insemination. Briefly, female B6D2F1 mice (Charles River Laboratories, Kanagawa, Japan), aged 9–14 weeks, were intraperitoneally injected with 5 IU of serum gonadotrophin (Serotropin®; ASKA Animal Health Co., Tokyo, Japan), followed by injection of 5 IU of human chorionic gonadotrophin (Gonatropin®; ASKA Animal Health Co.) after 48 h to induce ovulation. After an additional 14 h, female mice were euthanised through cervical dislocation. The oocytes were collected from these unmated mice and fertilised via IVF with ICR mouse sperm in HTF medium (Ark Resource, Kumamoto, Japan). Male mice were euthanised by cervical dislocation 1 h before insemination, and their spermatozoa were retrieved from the cauda epididymis by squeezing. The retrieved spermatozoa were suspended in HTF medium at 37˚C and 5% $CO_2$ for 1 h and then transferred to the IVF medium (at a final concentration of 200 sperm/µL).

## Embryo culture

Five hours after insemination, IVF embryos were transferred into KSOM (Ark Resource) and cultured in 200 µL of medium in a Primo Vision Embryo Culture Dish (Vitrolife, Gothenburg, Sweden), which was then covered with mineral oil. Embryos were cultured under the following conditions: 5% $O_2$, 5% $CO_2$, 90% $N_2$, and a temperature of 37˚C. Culture medium was not replaced during the experiments [15]. Embryos were imaged using the Primo Vision time-lapse monitoring system (Vitrolife) at 5-min intervals from IVF to vitrification, from thawing to experimental use, and from washing after experimental use with the electrolyte indicator for at least 24 h.

## Vitrification and thawing

Embryos were cultured *in vitro* until they reached the 2-cell stage, followed by vitrification using a Cryotop Safety Kit (Kitazato, Shizuoka, Japan). The embryos were immersed in equilibration solution for 15 min and then transferred into vitrification solution within 90 s. Next, they were placed on a Cryotop under a microscope. The Cryotop was frozen directly in liquid nitrogen, with 2–4 embryos vitrified in each Cryotop. For thawing, the embryos were warmed by submerging the Cryotop in thawing solution (1 mol/L sucrose) at 37˚C for <60 s and then in dilution solution (0.5 mol/L sucrose) for 3 min. The embryos were immersed in washing solution twice, for 5 and 1 min, respectively, washed three times in KSOM, and cultured.

## $Na^+/K^+$ ATPase functional assay using ouabain

Ouabain, a plant-derived cardiotonic steroid, binds to $Na^+/K^+$ ATPase and inhibits its ion transport function [16]. Ouabain octahydrate (Abcam, Cambridge, UK) was used to assess the function of $Na^+/K^+$ ATPase. Briefly, embryos were placed in $10^{-3}$ M ouabain for 1 h, the concentration previously reported to block the ion transport functions of $Na^+/K^+$ ATPase in mice [16, 17]. Subsequently, the embryos were loaded with electrolyte indicators.

## Electrolyte indicator

**Na⁺ imaging.** Embryos (from morula to hatching stage) were loaded with 0.5 μM CoroNa Green AM (Thermo Fisher Scientific, Waltham, MA, USA) according to the manufacturer's instructions and a previous report [12] with slight modifications, at 37°C for 30 min. Subsequently, intracellular and/or blastocoel fluorescence intensity was measured using an LSM780 confocal laser-scanning microscope (Carl Zeiss AG, Jena, Germany). The excitation and emission wavelengths were 488 and 510–520 nm, respectively (Fig 1A). The same experiments were repeated at least three times.

**K⁺ imaging.** Briefly, the embryos were loaded with 5 μM ION Potassium Green-2 AM (Abcam) at 37°C for 30 min according to a previous report [18]. The embryos were washed three times with KSOM, and the intracellular fluorescence intensity was measured. The excitation wavelength was set to 526 nm (6-JOE), and emission was detected at 540–560 nm (Fig 2A). The same experiments were repeated at least three times.

In total, 180 mouse embryos were observed in this study. For Na⁺ imaging, 96 embryos were studied, including 55 in the control group [morula (n = 20), blastocyst (n = 18), and hatching (n = 17)] and 41 in the ouabain group [morula (n = 17), blastocyst (n = 12), and hatching (n = 12)]. For K⁺ imaging, 84 embryos were used, including 50 in the control group [morula (n = 22), blastocyst (n = 11), and hatching (n = 17)] and 34 in the ouabain group [morula (n = 16), blastocyst (n = 9), and hatching (n = 9)].

## Intensity measurement

As these indicators have not been used for thick samples such as mouse embryos (height = 75–100 μm), we repeated these experiments to set optimal measurement conditions. The intensity was highest at the cross-section near the dish contact surface closest to the laser firing point and gradually decreased as the distance from the laser emission point increased. To compare the intensity of each embryo, measurement at almost the same height was conducted for all embryos. Images were taken at 5-μm intervals, and a slice containing the target cell or blastocoel was observed within 30 μm from the lowest plane of the embryo (near the equatorial plane of the embryo) (Fig 3A).

When measuring intracellular intensity, the outer cells were targeted at the morula stage, and the TE cells inside the zona pellucida were targeted after the blastocyst stage. The circumference of the all target cells was manually plotted in the same Z section (Fig 3B). In addition, polar body and degenerated cells showed high intensities; therefore, these cells were excluded from the target. Next, the mean intensity of the selected region was calculated using ZEN Software 2012 black edition (Carl Zeiss AG). The software measures intensity on a pixel-by-pixel basis, and automatically calculates the mean intensity (Fig 3C).

**Antibodies.** Embryos were stained with the following antibodies: goat polyclonal anti-Oct3/4 (Cat. Sc-8628, Lot #K0615; Santa Cruz Biotechnology, Dallas TX, USA) [15, 19], mouse monoclonal anti-Na⁺/K⁺-ATPase α-1 (Cat. #05–369, Lot 3170808; C464.6; Merck Millipore, Billerica, MA, USA) [15, 20], and Hoechst 33342 (Cat. 346–07951, Lot DH039, Dojindo Molecular Technologies, Kumamoto, Japan) [15].

## Immunofluorescence and confocal microscopy

For Oct3/4 and Na⁺/K⁺-ATPase staining, the embryos were fixed in methanol as previously recommended [21] with slight modification. The embryos were fixed for 3 min each in 2:1 PBS:methanol, 1:1 PBS:methanol, 1:2 PBS:methanol, and 100% methanol. They were then rehydrated using the same series in reverse. Prior to immunostaining, permeabilisation and blocking were performed using PHEM buffer (60 mM PIPES, 25 mM HEPES, 10 mM EGTA,

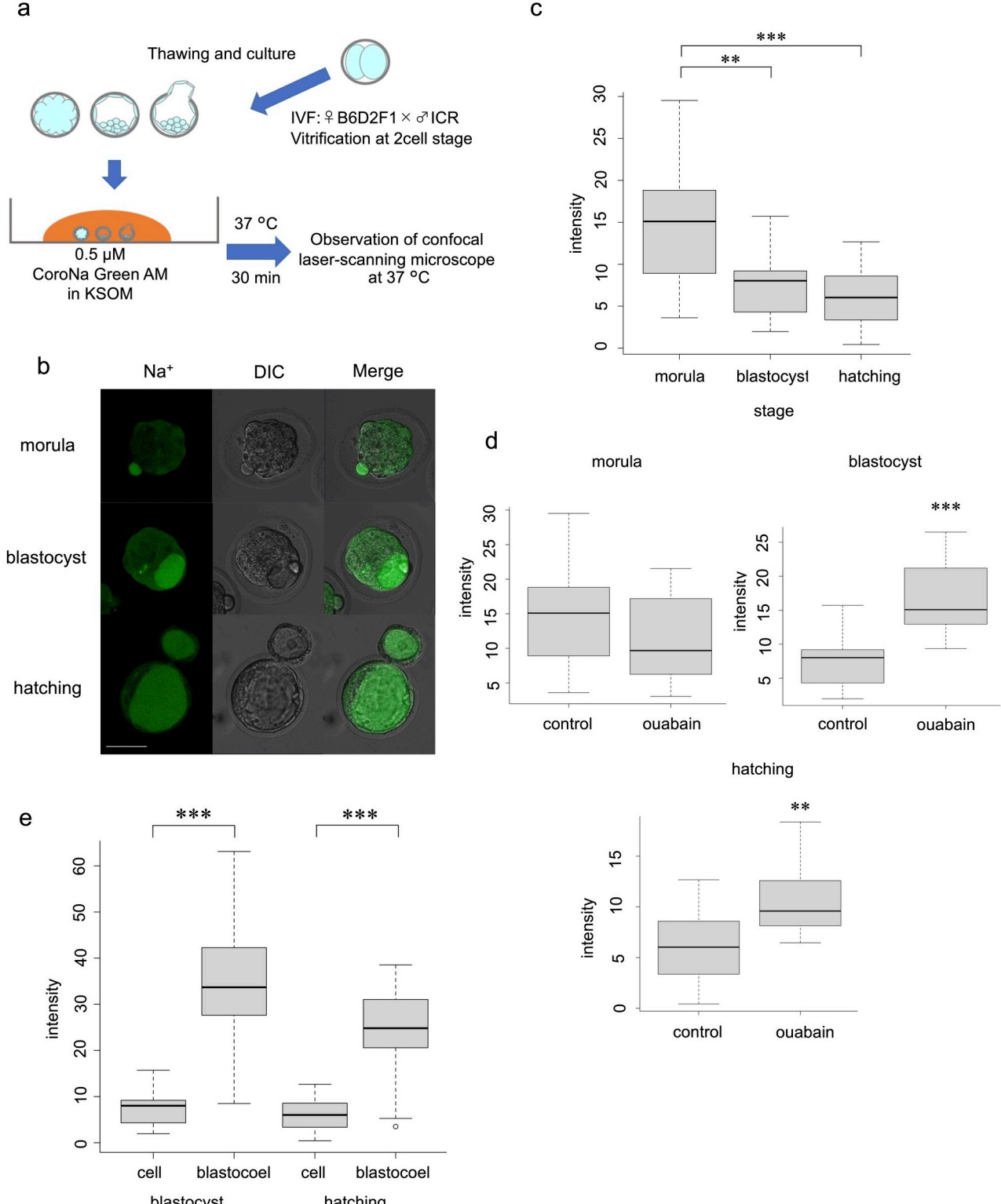

**Fig 1. Live visualisation of Na⁺ and intensity analysis using CoroNa Green AM.** (a) Protocol for CoroNa Green AM. (b) Images of embryos at morula, blastocyst, and hatching stages. Scale bar, 50 μm. (c) Comparison of CoroNa Green AM fluorescence intensity at each stage without ouabain. Wilcoxon rank sum test: $^{**}p < 0.01$, $^{***}p < 0.001$. (d) Effect of ouabain at each stage. Wilcoxon rank sum test: $^{**}p < 0.01$, $^{***}p < 0.001$. (e) Comparison of intracellular and blastocoel fluorescence intensity. Wilcoxon rank sum test: $^{***}p < 0.001$.

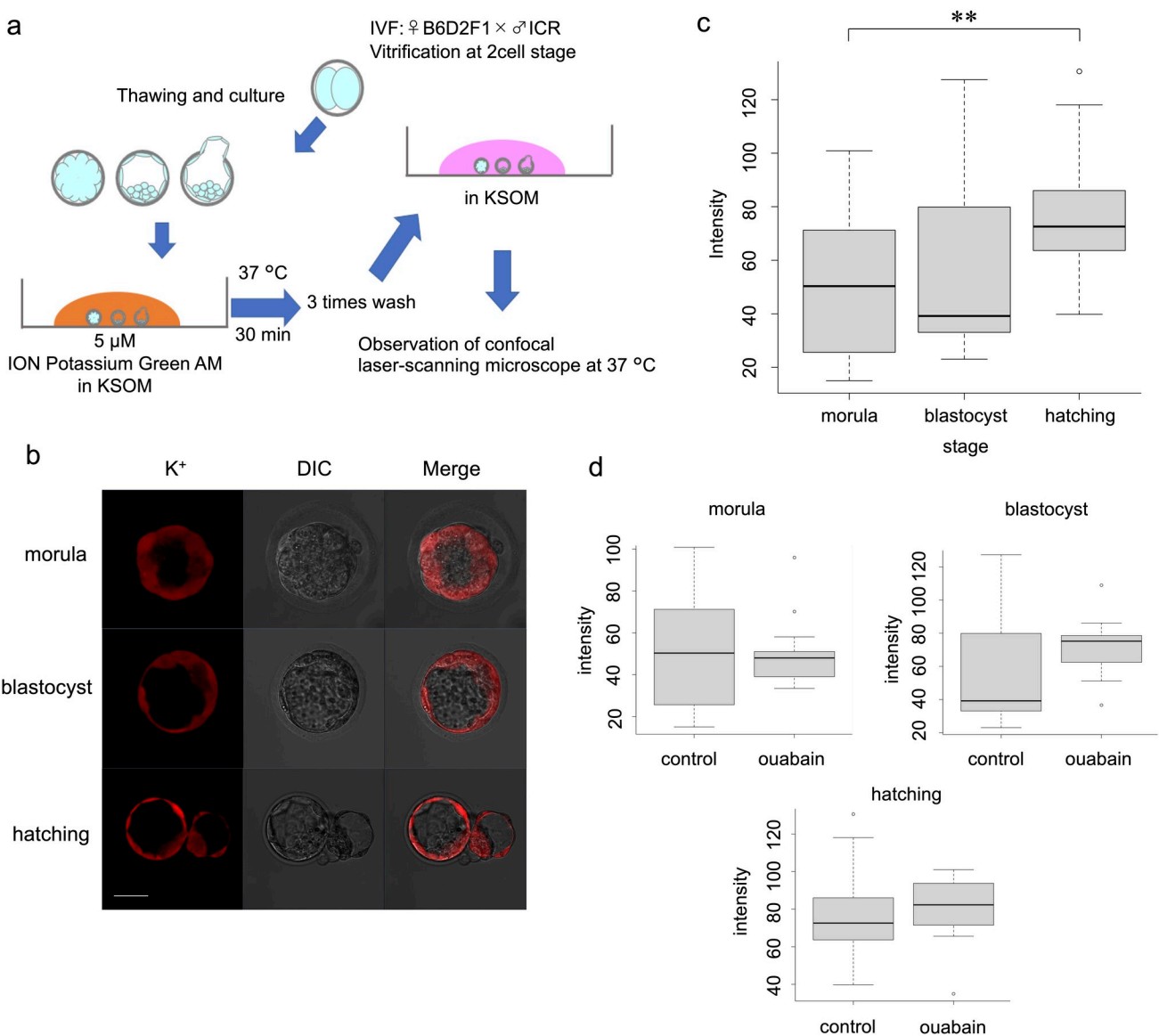

**Fig 2. Live visualisation of K⁺ and intensity analysis using ION Potassium Green-2 AM.** (a) Protocol for ION Potassium Green-2 AM. (b) Images at morula, blastocyst, and hatching stages. Scale bar, 50 μm. (c) Comparison of ION Potassium Green-2 AM fluorescence intensity at each stage without ouabain. Wilcoxon rank sum test: **p < 0.01. (d) Effect of ouabain at each stage.

2 mM MgCl$_2$ at pH 6.9) containing 0.01% Triton X-100 and 3% bovine serum albumin (BSA) for 45 min at room temperature. The embryos were washed three times in PBS containing 0.1% BSA, followed by incubation with a mixed solution of anti-Oct3/4 (1:50) and anti-Na+/K +-ATPase α-1 (1:200) primary antibodies overnight at 4˚C. The embryos were washed three times in wash buffer (PBS; Gibco® DPBS, Thermo Fisher Scientific, containing 0.1% BSA; Sigma-Aldrich, St. Louis, MO, USA) and incubated with secondary antibodies (1:200; Alexa Fluor 647-conjugated donkey anti-goat IgG, Abcam) for 1 h at room temperature. The embryos were again washed three times in wash buffer and incubated with secondary antibod-ies (1:200; Alexa Fluor 488-conjugated goat anti-mouse IgG, Abcam) for 1 h at room tempera-ture. Following another three washes in wash buffer, chromatin was stained with Hoechst

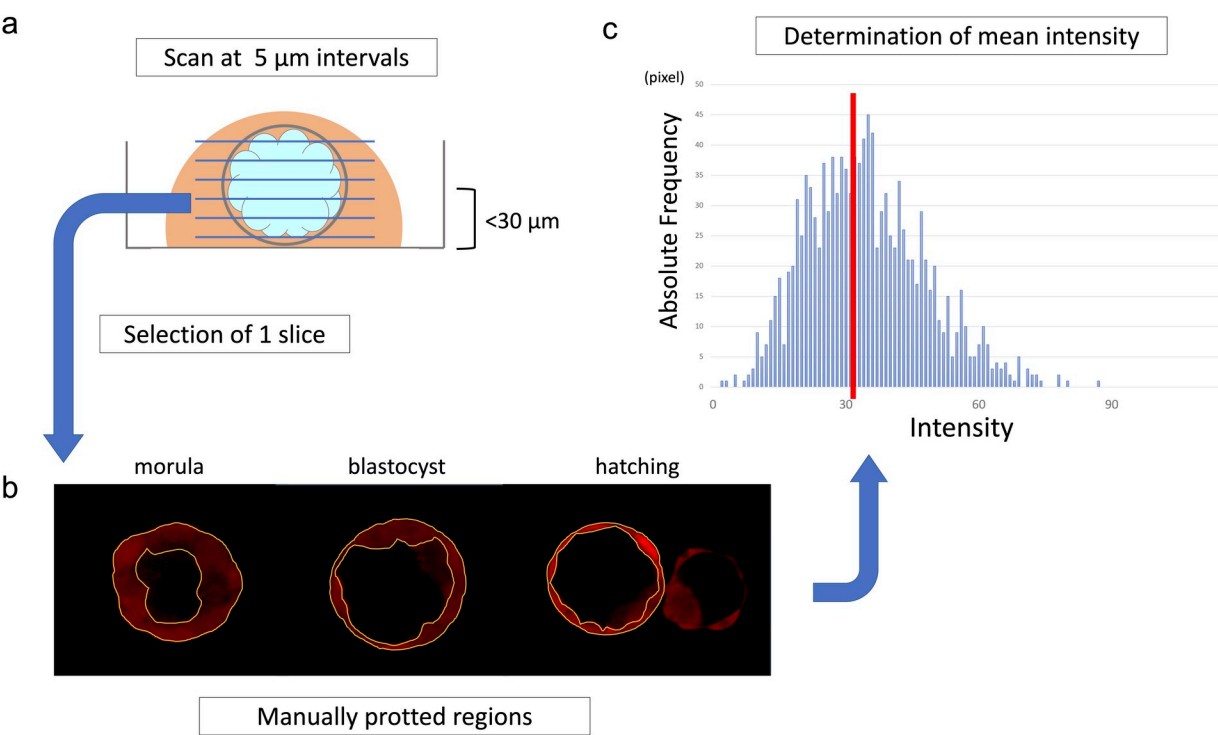

**Fig 3. Intensity measurement procedure.** (a) The image was taken at a slice thickness of 5 μm, and the optimum slice was selected within a range of 30 μm from the bottom. (b) The circumference of the target cells or the blastocoel was manually plotted to determine the measurement region. (c) The ZEN Software measured the intensity on a pixel-by-pixel basis and automatically calculated the mean intensity.

33342 (1:1000) for 20 min at room temperature. After washing three times in wash buffer, the stained embryos were examined under an LSM780 confocal laser-scanning microscope (Carl Zeiss AG).

## Statistical analyses

All statistical analyses were performed using The R Project for Statistical Computing (R Version 4.0.3, Vienna, Austria). The percentages of normal growth in the control group and test groups after using electrolyte indicators with/without ouabain were compared using Fisher's exact test and post-hoc pairwise comparisons adjusted by Holm's method. Wilcoxon rank sum test was used to compare fluorescence intensities between the control and ouabain groups at each stage. Wilcoxon signed rank exact test was used to compare fluorescence intensities between cells and blastocoels at the blastocyst and hatching stages, and intensities between TE-blastocoel and TE-ICM. Fluorescence intensities between the embryo stages were compared using a nonparametric, pairwise, multiple-comparison procedure, followed by Kruskal–Wallis or Dunn's tests [22]. $p < 0.05$ was considered to indicate statistical significance.

## Results

### Effect of electrolyte indicator on the mouse embryo

After loading the electrolyte indicator, the growth of the embryos was examined using the Primo Vision time-lapse monitoring system for at least 24 h after washing three times. Briefly, 94.9% (37/39) of the embryos in the control group, without the indicator and ouabain, 92.0% (23/25) in the CoroNa Green AM group, 91.9% (34/37) in the ouabain + CoroNa Green AM

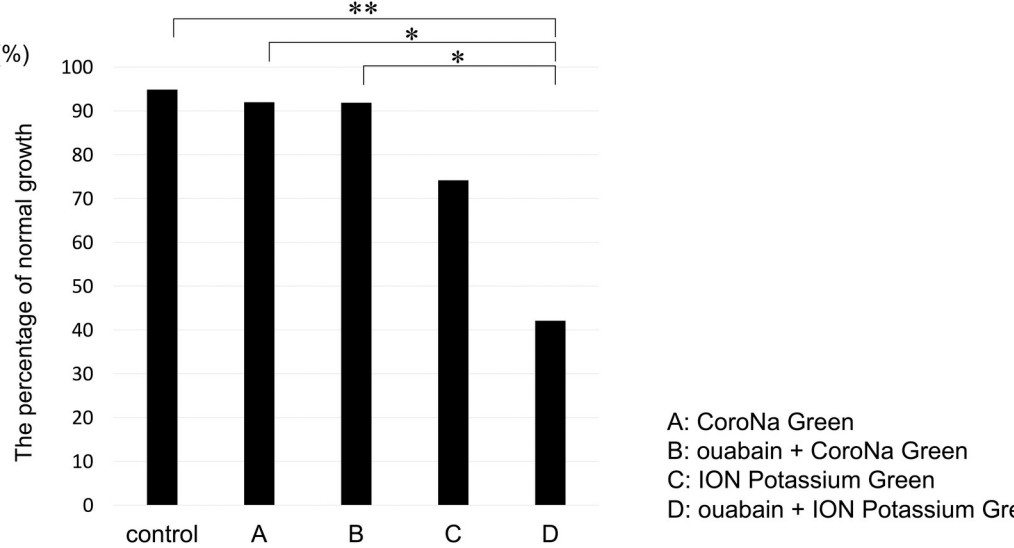

**Fig 4. Comparison of normal growth after using electrolyte indicators with/without ouabain.** (A) CoroNa Green AM (B) ouabain + CoroNa Green AM (C) ION Potassium Green-2 AM (D) ouabain + ION Potassium Green-2 AM. Normal growth was 94.9% (37/39) in control, 92.0% (23/25) in (A), 91.9% (34/37) in (B), 74.2% (23/31) in (C), and 42.1% (16/38) in (D). Fisher's exact test and post-hoc pairwise comparisons adjusted using Holm's method: *p < 0.05, **p < 0.01.

group, 74.2% (23/31) in the ION Potassium Green-2 AM group, and 42.1% (16/38) in the ouabain + ION Potassium Green-2 AM group showed normal growth. Therefore, the ouabain + ION Potassium Green-2 AM group was significantly affected compared to the control (p = 0.0040), CoroNa Green AM (p = 0.041), and ouabain + CoroNa Green AM (p = 0.0192) groups (Fig 4), indicating that ION Potassium Green-2 AM was more cytotoxic than CoroNa Green AM, particularly in the ouabain combination group. (p = 1.000, control vs. CoroNa Green and ouabain + CoroNa Green groups; p = 0.129, control vs. ION Potassium Green) (Fig 4)

## Na⁺ imaging

In all embryonic developmental stages, the cytoplasm exhibited almost uniform fluorescence (Fig 1B). The change in intensity over time after addition of CoroNa Green AM gradually increased and reached a plateau after 20 min, indicating that the indicator was well dispersed and de-esterified in the embryo after reaching a plateau. During time-lapse imaging of the embryo after the blastocyst stage, the outer cells initially showed fluorescence, followed immediately by the blastocoel (S1 Movie). This result indicates that CoroNa Green AM is excreted into the blastocoel after entering the cell.

### Comparison of CoroNa Green AM fluorescence intracellular intensities at each stage

The blastocyst stage and the hatching stage showed significantly lower fluorescence intensities than the morula stage (p = 0.0062, morula vs. blastocyst; p < 0.001, morula vs. hatching group). The difference between the blastocyst stage and the hatching stage was not significant (p = 0.3700) (Fig 1C). Imaging was performed by loading the embryos with the Na⁺/K⁺ ATPase inhibitor ouabain, which induced a significant increase in fluorescence intensity in the blastocyst group (p < 0.001) and in the hatching stage (p = 0.0043). Ouabain loading did not induce a significant difference in fluorescence in the morula group (p = 0.1373) (Fig 1D).

## Comparison of intracellular and blastocoel CoroNa Green AM fluorescence intensities

After the blastocyst stage, the fluorescence intensity of the blastocoel cavity was significantly higher than that of the intracellular environment at both the blastocyst and hatching stages ($p < 0.001$) (Fig 1E).

## $K^+$ imaging

The results of $K^+$ and $Na^+$ imaging differed. Time-lapse imaging after addition of ION Potassium Green-2 AM revealed an immediate gradual increase in intensity (S2 Movie), which did not reach a plateau thereafter. Cells that were not in direct contact with the reagent, such as inner cells of the morula and ICM, showed no or very weak fluorescence (Fig 2B and S2 Movie).

## Comparison of ION Potassium Green-2 AM fluorescence intracellular intensities at each stage

The hatching group showed significantly higher fluorescence intensity than that of the morula group ($p = 0.0072$). The fluorescence intensity did not differ significantly between the blastocyst group and the preceding and following stages ($p = 0.9269$, morula vs. blastocyst; $p = 0.0684$, blastocyst vs. hatching group). However, intracellular fluorescence intensity increased as development progressed in the morula, blastocyst, and hatching groups (Fig 2C). Loading embryos with ouabain did not induce a significant difference in any group (control vs. ouabain, $p = 0.9374$ at the morula stage; $p = 0.1479$ at the blastocyst stage; $p = 0.5225$ at the hatching stage) (Fig 2D).

## Comparison of fluorescence intensities of ION Potassium Green-2 AM between TE-blastocoel and TE-ICM

After blastocoel formation, TE cells were divided into cells directly in contact with the blastocoel (TE-blastocoel) and ICM (TE-ICM) (Fig 5B). The fluorescence intensities in the TE-blastocoel revealed significantly higher intensity than in TE-ICM ($p < 0.001$) (Fig 5A and 5C). Additionally, immunostaining confirmed that $Na^+/K^+$ ATPase was localised and expressed on the basolateral cell membranes of TE-ICM, similar to that in the TE-blastocoel (Fig 5D).

## Discussion

To our knowledge, this is the first study using CoroNa Green AM and ION Potassium Green-2 AM indicators to visualise intracellular electrolytes during embryonic development of the mouse embryo.

We observed the behaviour of electrolytes in blastocoel formation. Our results support the previously reported function of $Na^+/K^+$ ATPase; we additionally observed changes in the intracellular electrolyte concentration. Changes in the intensities of $Na^+$ and $K^+$ from morula to hatching demonstrated that the intracellular $Na^+$ concentration decreased while $K^+$ concentration increased during blastocoel formation. This result supports the function of $Na^+/K^+$ ATPase shown in many previous studies [7–9]. When ouabain was used in combination with $Na^+$ imaging, the intensity increased significantly in the blastocyst and hatching groups. These results demonstrate the increase in $Na^+$ concentration due to $Na^+/K^+$ ATPase inhibition, and we considered CoroNa Green AM imaging to correlate with the intracellular $Na^+$ concentration. In contrast, at the morula stage, the intensity was not affected by ouabain, because the activity of $Na^+/K^+$ ATPase was lower than that at the blastocyst stage [23]. Additionally, we

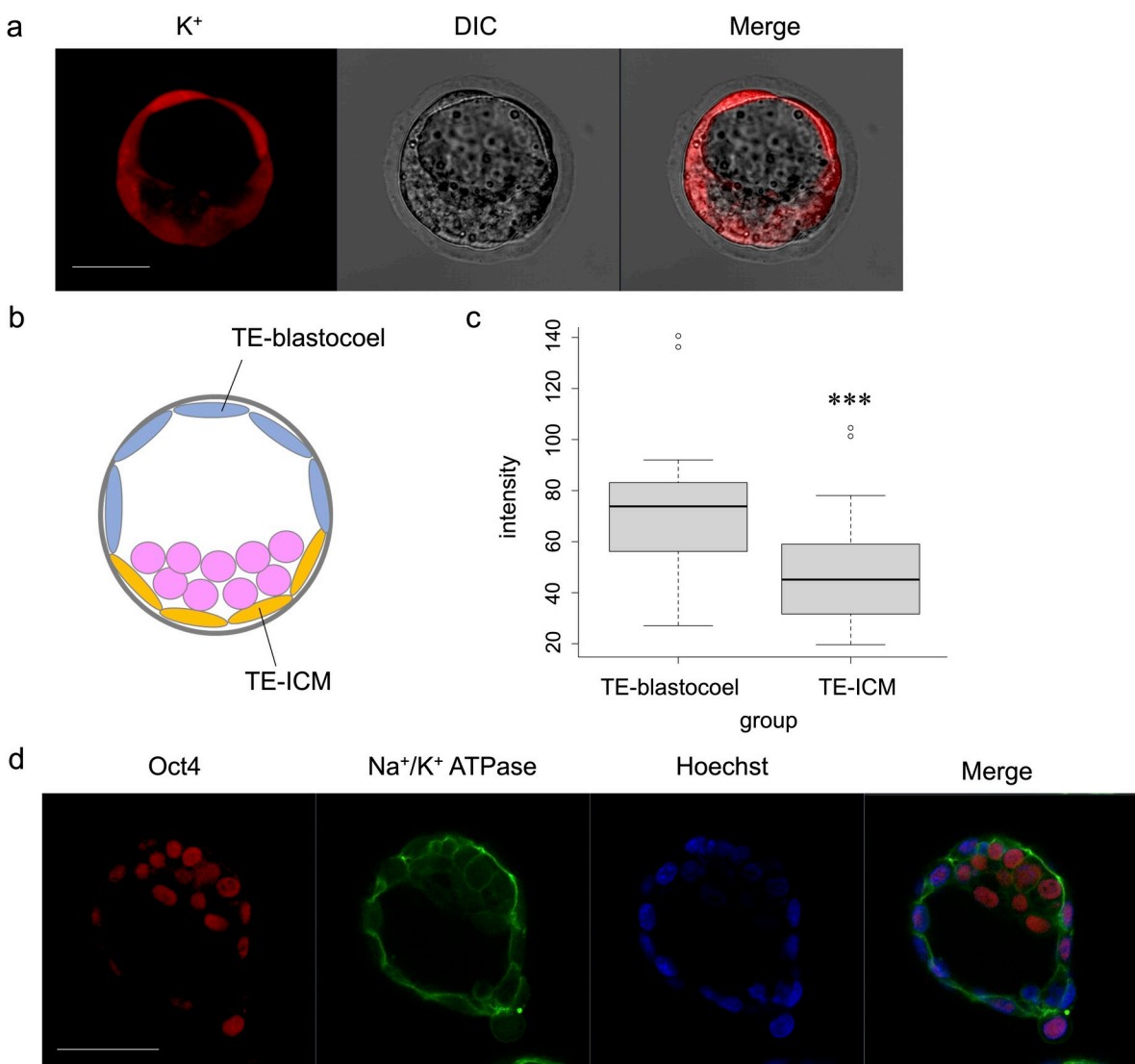

**Fig 5. Comparison of ION Potassium Green-2 AM fluorescence intensity between TE-blastocoel and TE-ICM.** (a) Imaging of ION Potassium Green-2 AM at the blastocyst stage. Scale bar, 50 μm. (b) Schematic illustration of TE-blastocoel and TE-ICM. TE is divided into cells directly in contact with the blastocoel (TE-blastocoel) and ICM (TE-ICM). (c) Comparison of fluorescence intensity between TE-blastocoel and TE-ICM. Wilcoxon signed rank exact test: ***p < 0.001. (d) Immunofluorescence images (Oct3/4, Na⁺/K⁺ ATPase, Hoechst, Merged) at the hatching stage. Scale bar, 50 μm.

thought that the activity of $Na^+/K^+$ ATPase was originally suppressed at the morula stage, which may have contributed to $Na^+$ storage in TE cells for blastocoel formation. We found that a greater change in intensity after using ouabain was associated with higher activity of $Na^+/K^+$ ATPase, which is consistent with a study that reported the highest $Na^+/K^+$ ATPase activity in the blastocyst [23]. Notably, $K^+$ fluorescence intensity did not change with ouabain loading. This may be attributed to the strong cytotoxicity observed in the ION Potassium Green-2 and ouabain combination group, and thus accurate measurements could not be performed.

Intensity in the blastocoel was higher than that in the cell, indicating that the blastocoel had a high $Na^+$ concentration, as previously suggested [7–9]. This is the first study to visualise high

Na$^+$ inside the blastocoel. However, it is unclear whether this was due to a difference in the indicator concentration between cells and the blastocoel and requires further analysis.

K$^+$ intensity was higher in the TE-blastocoel than in the TE-ICM, indicating that the activity of Na$^+$/K$^+$ ATPase differed depending on its location in the TE (Fig 5A and 5C). When Na$^+$/K$^+$ ATPase expression was confirmed by immunostaining, it was similarly localized to the basolateral cell membrane of TE-blastocoel and TE-ICM (Fig 5D). This indicates that the electrolyte indicator can reveal differences in pump activity that cannot be visualised by immunostaining. Low Na$^+$/K$^+$ ATPase activity in the TE-ICM corresponded with embryonic behaviour during blastocoel formation, supporting that changes in electrolyte concentrations were visualised.

Unlike time-lapse K$^+$ imaging, fluorescence intensity reached a plateau in time-lapse Na$^+$ imaging because CoroNa Green AM was more easily excreted from the cells than was ION Potassium Green-2 AM. It has been reported that CoroNa Green AM leaks out of cells more easily than other Na$^+$ indicators [13]. Although a multidrug resistance protein is reportedly involved in the extracellular emission of fluorescent cellular indicators [24, 25], details regarding the two indicators used in this study are unclear. In addition, the lower percentage of embryos showing normal development in the ION Potassium Green-2 AM group compared to in the CoroNa Green AM group may be related to the low degree of excretion of ION Potassium Green-2 AM indicator from the cells.

In this study, CoroNa Green AM and ION Potassium Green-2 AM were successfully used for the first time in mouse embryos to visualise the electrolyte concentration in real-time. Based on these results, new findings were revealed regarding the transition of Na$^+$/K$^+$ ATPase activity at each stage in blastocoel formation and differences in Na$^+$/K$^+$ ATPase activity depending on the location of TE. This method will enable observation of the effects of electrolyte concentrations of various culture solutions within embryos which could not be clarified until now. We hope that this will lead to the development of improved culture solutions and refinements to culture environments in the future, which will greatly contribute to developments in reproductive medicine.

## Supporting information

**S1 File.**
(PDF)

**S1 Movie. Time-lapse imaging with CoroNa Green AM.** This movie was created by editing time-lapse images obtained at 30 seconds intervals from 3 to 40 minutes after adding CoroNa Green AM.
(MP4)

**S2 Movie. Time-lapse imaging with ION Potassium Green-2 AM.** This movie was created by editing a time-lapse image obtained at 30 seconds intervals from 4 to 40 minutes, after adding ION Potassium Green-2 AM.
(MP4)

## Acknowledgments

We would like to acknowledge our team for their role in completing this research.

We would also like to thank Editage [http://www.editage.com] for editing this manuscript for English language.

## Author Contributions

**Conceptualization:** Akiko Fujishima, Kazumasa Takahashi, Yukihiro Terada.

**Data curation:** Akiko Fujishima, Kazumasa Takahashi, Mayumi Goto.

**Formal analysis:** Akiko Fujishima, Eri Maeda.

**Funding acquisition:** Yukihiro Terada.

**Investigation:** Akiko Fujishima, Kazumasa Takahashi, Mayumi Goto.

**Methodology:** Akiko Fujishima, Kazumasa Takahashi.

**Project administration:** Akiko Fujishima, Kazumasa Takahashi.

**Resources:** Akiko Fujishima, Kazumasa Takahashi, Mayumi Goto.

**Software:** Akiko Fujishima, Mayumi Goto.

**Supervision:** Kazumasa Takahashi, Hiromitsu Shirasawa, Yukiyo Kumazawa, Yukihiro Terada.

**Validation:** Akiko Fujishima, Kazumasa Takahashi, Mayumi Goto.

**Visualization:** Akiko Fujishima, Mayumi Goto.

**Writing – original draft:** Akiko Fujishima, Yukihiro Terada.

**Writing – review & editing:** Akiko Fujishima, Kazumasa Takahashi, Takeo Hirakawa, Takuya Iwasawa, Kazue Togashi, Eri Maeda, Hiromitsu Shirasawa, Hiroshi Miura, Wataru Sato, Yukiyo Kumazawa, Yukihiro Terada.

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
