## [Decision Letter · Decision Letter 0]

4 Nov 2020

PONE-D-20-29923

Live visualisation of electrolytes during mouse embryonic development using electrolyte indicators

PLOS ONE

Dear Dr. Fujishima,

Thank you for submitting your manuscript to PLOS ONE. After careful consideration, we feel that it has merit but does not fully meet PLOS ONE’s publication criteria as it currently stands. Therefore, we invite you to submit a revised version of the manuscript that addresses the points raised during the review process.

We look forward to receiving your revised manuscript.

Kind regards,

Peter J. Hansen

Academic Editor

PLOS ONE

Journal Requirements:

"This work was supported by JSPS KAKENHI, Grant-in-Aid for Scientific Research (B) [Grant Number 18H02942].".

i) Please provide an amended statement that declares *all* the funding or sources of support (whether external or internal to your organization) received during this study, as detailed online in our guide for authors at http://journals.plos.org/plosone/s/submit-now.  Please also include the statement “There was no additional external funding received for this study.” in your updated Funding Statement.

ii) Please include your amended Funding Statement within your cover letter. We will change the online submission form on your behalf.

Reviewers' comments:

Reviewer's Responses to Questions

**Comments to the Author**

1. Is the manuscript technically sound, and do the data support the conclusions?

Reviewer #1: Partly

2. Has the statistical analysis been performed appropriately and rigorously? 

Reviewer #1: Yes

3. Have the authors made all data underlying the findings in their manuscript fully available?

Reviewer #1: No

4. Is the manuscript presented in an intelligible fashion and written in standard English?

Reviewer #1: No

5. Review Comments to the Author

Reviewer #1: In this manuscript, written by Akiko Fujishima and colleagues, the authors study the location of Na+ and K+ electrolytes during mouse preimplantation development. They quantify the electrolytes presence and changes during development by time-lapse confocal microscopy thanks to specific indicators (CoroNa Green AM and ION Potassium Green-2 AM respectively for Na+ and K+). The study focus between morula and late blastocyst (hatching stage) stages, when first lineages and cavitation appear. They show different intensity of the electrolytes indicators depending of the developmental stages, of the cell type (trophectoderm versus inner cell mass) and in the blastocoel. It is of interest for the community to better know Potassium and Sodium concentration in intracellular blastomeres as well as in the blastocoel in this crucial developmental period. However, I have several caveats in the way the analysis is performed and I can not support publication without major reviewing of the manuscript.

1) All the experiments are performed on frozen IVF embryos. It is quite surprising when both IVF and freezing could impact embryonic development and lead to lower developmental yield compared to natural mating. The authors should argument why they did not use superovulated mice followed by natural mating.

2) The authors should avoid overstatement in their abstract and modify the text accordingly. This study does not “confirm the function of Na+/K+ ATPase described in previous studies” (line 37).

3) The authors claim that they have performed time-lapse imaging between zygote and 2-cell stage (why if not in presence of the electrolyte indicators?) and after thawing but none of these data are provided in the manuscript. If time-lapse data are available during blastocoel formation and hatching, it is important to share the movies as well as the quantification of intensity changes over time during these important processes.

4) How the intensity quantification has been performed should be better explain in the Methods section. What are the three replicates that the authors have performed to set up the optimal measurement conditions and what were their controls? A scheme with their imaging features and step-by-step image analysis would be highly appreciated as most of their results are based on this intensity measurement.

How many cells per embryo are analysed? The intensity should be normalized to the size of the cell (or blastocoel) in the Z section where the analysis is performed.

5) The low level of embryonic development in presence of 5 uM ION Potassium Green-2 AM (74%), worsen with added ouabain (42%), is problematic and shows cytotoxicity for blastocyst development. Different concentration of the Potassium electrolyte indicator should be tested to find a more suitable concentration and avoid confounding effect.

6) Line 282, “This result indicates that CoroNa Green AM was excreted into the blastocoel after entering the cell”. This results would be better supported by sharing the performed time-lapse movies and analysing the changes over time for the same embryo.

6. PLOS authors have the option to publish the peer review history of their article (what does this mean?). If published, this will include your full peer review and any attached files.

Reviewer #1: **Yes: **Maud Borensztein

---

## [Author Response · Author response to Decision Letter 0]

19 Nov 2020

To Reviewer 1

Thank you for the thoughtful and constructive feedback for our manuscript titled “Live visualisation of electrolytes during mouse embryonic development using electrolyte indicators.” 

1) All the experiments are performed on frozen IVF embryos. It is quite surprising when both IVF and freezing could impact embryonic development and lead to lower developmental yield compared to natural mating. The authors should argument why they did not use superovulated mice followed by natural mating.

Response: Thank you for the valuable suggestion. We used IVF because we could accurately identify the fertilization time, obtain more embryos, and make the culture environment uniform from the time of insemination. We have added the sentence “and maintain a uniform developmental environment throughout the experimental period, starting from the time of insemination.” to our manuscript (Collection of mouse oocytes, zygotes, and embryos in Material and Methods, page 7, lines 106-108). We used frozen embryos to efficiently obtain the embryos in the target developmental stage for observation using confocal laser-scanning microscope. In addition, after thawing, recovery culture was performed until the embryos reached the target stage. The embryos were screened using time-lapse observation, and those that exhibited abnormal development were excluded. Moreover, we think that the effects of vitrification and thawing are not significant in this experiment, considering that the embryos were subjected to sufficient recovery culture and that abnormal embryos were excluded.

2) The authors should avoid overstatement in their abstract and modify the text accordingly. This study does not “confirm the function of Na+/K+ ATPase described in previous studies” (line 37).

Response: Thank you for pointing this out. We have rewritten the text as follows: “The changes in electrolyte concentration observed in this study were consistent with the activity of Na+/K+ ATPase reported previously” (Abstract, page 3, lines 37-39)

3) The authors claim that they have performed time-lapse imaging between zygote and 2-cell stage (why if not in presence of the electrolyte indicators?) and after thawing but none of these data are provided in the manuscript. If time-lapse data are available during blastocoel formation and hatching, it is important to share the movies as well as the quantification of intensity changes over time during these important processes.

Response: Thank you for your valuable insights. We used time-lapse monitoring from zygote to 2-cell stage to screen for normally developing embryos. We used the Primo Vision time-lapse monitoring system, and not confocal laser-scanning microscopy. Due to the duplication of information pertaining to time-lapse imaging with Primo Vision and confocal-laser scanning microscopy, we have removed the paragraph "Time-lapse imaging" in Material and Methods. We have rewritten the "time-lapse imaging" description confined only to confocal laser-scanning microscopy. However, time-lapse images obtained using confocal laser-scanning microscopy were not used for analysis, because frequent laser irradiation causes fading of the indicator and would be unsuitable for comparing the intensities accurately. In this study, the indicator was added to embryos that had grown to the target stage, and the images obtained through a single scan were used for analysis.

4) How the intensity quantification has been performed should be better explain in the Methods section. What are the three replicates that the authors have performed to set up the optimal measurement conditions and what were their controls? A scheme with their imaging features and step-by-step image analysis would be highly appreciated as most of their results are based on this intensity measurement.

How many cells per embryo are analyzed? The intensity should be normalized to the size of the cell (or blastocoel) in the Z section where the analysis is performed

Response: We apologize for the lack of clarity. We have added the details of the experimental procedure, and added a figure (Fig 3) to the Materials and Methods, under the subheading Intensity measurement (page 12, 13, line 189-211). We analyzed one cell from each embryo at each developmental stage to measure intracellular intensity, and the detailed procedure is described in the manuscript. 

We have also added, “The same experiments were repeated at least three times.” to our manuscript (Electrolyte indicator, page 10, line 167, and page 11, line 187).

5) The low level of embryonic development in presence of 5 μM ION Potassium Green-2 AM (74%), worsen with added ouabain (42%), is problematic and shows cytotoxicity for blastocyst development. Different concentration of the Potassium electrolyte indicator should be tested to find a more suitable concentration and avoid confounding effect.

Response: Thank you for your valuable comment. The appropriate concentration of ION Potassium Green-2 AM was not mentioned in the manufacturer’s instructions, and there are only a few reports for reference. Therefore, we used the concentration stated in a previous report (Sekar P, Oncotarget. 2018). In order to investigate the characteristics of ION Potassium Green-2 AM, time-lapse imaging was performed. As a result, the optimum loading time was determined. Additionally, in the time-lapse imaging, the intracellular intensity continued to increase, and the intensity remained stable even after washing and long-term culturing in medium without the indicator. It was predicted that ION Potassium Green-2 AM is difficult to be excreted, and this property is believed to be associated with its cytotoxicity. Therefore, we concluded that the effect of ION Potassium Green-2 AM on the embryo could not be eliminated even if the reagent concentration was reduced.

We have added the time-lapse movie with ION Potassium Green-2 AM as Supporting Information (S2 Movie).

6) Line 282, “This result indicates that CoroNa Green AM was excreted into the blastocoel after entering the cell”. This results would be better supported by sharing the performed time-lapse movies and analyzing the changes over time for the same embryo.

Response: We apologize for the gap in the information. Time-lapse images were not used for data analysis, but for understanding the characteristics of the CoroNa Green AM. During imaging, we found that the TE cell was visible initially followed by the blastocoel cavity. We have added the time-lapse movie with CoroNa Green AM as Supporting information (S1 Movie). 

Additional corrections

The following minor changes have been made:

The manuscript has been thoroughly proofread and English language issues have been corrected.

---

## [Decision Letter · Decision Letter 1]

31 Dec 2020

PONE-D-20-29923R1

Live visualisation of electrolytes during mouse embryonic development using electrolyte indicators

PLOS ONE

Dear Dr. Fujishima,

Thank you for submitting your manuscript to PLOS ONE. After careful consideration, we feel that it has merit but does not fully meet PLOS ONE’s publication criteria as it currently stands. Therefore, we invite you to submit a revised version of the manuscript that addresses the points raised during the review process.

We look forward to receiving your revised manuscript.

Kind regards,

Peter J. Hansen

Academic Editor

PLOS ONE

Reviewers' comments:

Reviewer's Responses to Questions

**Comments to the Author**

1. If the authors have adequately addressed your comments raised in a previous round of review and you feel that this manuscript is now acceptable for publication, you may indicate that here to bypass the “Comments to the Author” section, enter your conflict of interest statement in the “Confidential to Editor” section, and submit your "Accept" recommendation.

Reviewer #1: (No Response)

2. Is the manuscript technically sound, and do the data support the conclusions?

Reviewer #1: Yes

3. Has the statistical analysis been performed appropriately and rigorously? 

Reviewer #1: Yes

4. Have the authors made all data underlying the findings in their manuscript fully available?

Reviewer #1: Yes

5. Is the manuscript presented in an intelligible fashion and written in standard English?

Reviewer #1: Yes

6. Review Comments to the Author

Reviewer #1: Dear authors,

Thank you for improving your manuscript and in particular for the additional figure 3 and movies S1 and S2. However, I have still two points which need to be answered, including my former question 4 on intensity normalization.

1) According to your answers, I am surprised of your choice of analysing only one cell per embryo. Could you explain why you chose this strategy, please? The number of cells analysed could have been improved by studying at least all the cells of the same Z section or all the cells in contact with the reagents for the Potassium Green-2 AM.

2) In the first round of review, I have asked about the normalization of the mean intensity. According to your answer, it does not seem that you are normalizing the mean intensity by the size of the cell, which is very important to be able to compare your results.

Please, provide the normalised analysis of your results.

Best regards

7. PLOS authors have the option to publish the peer review history of their article (what does this mean?). If published, this will include your full peer review and any attached files.

Reviewer #1: **Yes: **Maud Borensztein

---

## [Author Response · Author response to Decision Letter 1]

12 Jan 2021

Thank you for the thoughtful and constructive feedback on our manuscript titled “Live visualisation of electrolytes during mouse embryonic development using electrolyte indicators.” The manuscript ID is PONE-D-20-29923.

To Reviewer 1

1) According to your answers, I am surprised of your choice of analysing only one cell per embryo. Could you explain why you chose this strategy, please? The number of cells analysed could have been improved by studying at least all the cells of the same Z section or all the cells in contact with the reagents for the Potassium Green-2 AM.

Response: Thank you for the valuable suggestion. As you pointed out, there was concern that selecting a single cell would bias the selection and would not provide accurate data. Therefore, we selected all the target cells of the same Z section and reanalyzed the data. However, it did not change much from the previous result. Regarding the comparison of CoroNa Green AM fluorescence intracellular intensities, in the previous result only the blastocyst stage showed a significant difference when compared to the morula stage, but in the new analysis, the blastocyst stage and the hatching stage showed significantly lower fluorescence intensities than the morula stage. Additionally, the ouabain and control groups showed significant differences not only in the blastocyst stage, but also in the hatching stage. 

2) In the first round of review, I have asked about the normalization of the mean intensity. According to your answer, it does not seem that you are normalizing the mean intensity by the size of the cell, which is very important to be able to compare your results.

Please, provide the normalized analysis of your results.

Response:

The ZEN software 2012 black edition, that was used in this study, measures intensity on a pixel-by-pixel basis, and automatically calculates the mean intensity and the number of pixels. Therefore, these intensities are not an area-dependent value. In addition, we searched for reports that measured fluorescence intensity, but could not find one that had normalized by the size of the measured area [1-4]. Since the used mean intensities are already normalized by the number of pixels (based on the reason mentioned previously), we think that their comparison is a valid approach

1. Oksana Iamshanova, Pascal Mariot, V'yacheslav Lehen'kyi, Natalia Prevarskaya. Comparison of fluorescence probes for intracellular sodium imaging in prostate cancer cell lines. Eur Biophys J.2016;45(7):765-777. doi: 10.1007/s00249-016-1173-7. Epub 2016 Sep 22.

2. Charitha Galva, Pablo Artigas, Craig Gatto. Nuclear Na+/K+-ATPase plays an active role in nucleoplasmic Ca2+ homeostasis. J Cell Sci. 2012;125(Pt 24):6137-47. doi: 10.1242/jcs.114959.

3. Jiantao Zhang, Hua Liu, Jian Sun, Bei Li, Qiang Zhu, Shaoliang Chen, et al. Arabidopsis fatty acid desaturase FAD2 is required for salt tolerance during seed germination and early seedling growth. PLoS One. 2012;7(1):e30355. doi: 10.1371/journal.pone.0030355.

4. Dong-Ha Oh, Eduardo Leidi, Quan Zhang, Sung-Min Hwang, Youzhi Li, Francisco J Quintero, et al. Loss of halophytism by interference with SOS1 expression. Plant Physiol. 2009;151(1):210-22. doi: 10.1104/pp.109.137802.

---

## [Decision Letter · Decision Letter 2]

19 Jan 2021

Live visualisation of electrolytes during mouse embryonic development using electrolyte indicators

PONE-D-20-29923R2

Dear Dr. Fujishima,

We’re pleased to inform you that your manuscript has been judged scientifically suitable for publication and will be formally accepted for publication once it meets all outstanding technical requirements.

Kind regards,

Peter J. Hansen

Academic Editor

PLOS ONE

Additional Editor Comments (optional):

Reviewers' comments:

Reviewer's Responses to Questions

**Comments to the Author**

1. If the authors have adequately addressed your comments raised in a previous round of review and you feel that this manuscript is now acceptable for publication, you may indicate that here to bypass the “Comments to the Author” section, enter your conflict of interest statement in the “Confidential to Editor” section, and submit your "Accept" recommendation.

Reviewer #1: All comments have been addressed

2. Is the manuscript technically sound, and do the data support the conclusions?

Reviewer #1: (No Response)

3. Has the statistical analysis been performed appropriately and rigorously? 

Reviewer #1: (No Response)

4. Have the authors made all data underlying the findings in their manuscript fully available?

Reviewer #1: (No Response)

5. Is the manuscript presented in an intelligible fashion and written in standard English?

Reviewer #1: (No Response)

6. Review Comments to the Author

Reviewer #1: (No Response)

7. PLOS authors have the option to publish the peer review history of their article (what does this mean?). If published, this will include your full peer review and any attached files.

Reviewer #1: **Yes: **Maud Borensztein

---

## [Editor Report · Acceptance letter]

21 Jan 2021

PONE-D-20-29923R2 

Live visualisation of electrolytes during mouse embryonic development using electrolyte indicators 

Dear Dr. Fujishima:

I'm pleased to inform you that your manuscript has been deemed suitable for publication in PLOS ONE. Congratulations! Your manuscript is now with our production department. 

Kind regards, 

on behalf of

Dr. Peter J. Hansen 

Academic Editor

PLOS ONE